# Magnetic Isolation of the Linear Trinuclear Anion in [Cu(Him)_6_] {Cu(Him)_4_[Cu(μ-EDTA)(Him)]_2_}·6H_2_O (1) as the Novel Imidazolium(+) Salt (H_2_im)_2_[Cu(Him)_4_{(µ-EDTA)Cu(Him)}_2_]·2H_2_O (2)—A Comparative Look to Their Crystal Structures, Thermal, Spectral and Magnetic Properties and DFT Calculations

**DOI:** 10.3390/ijms252313130

**Published:** 2024-12-06

**Authors:** Jeannette Carolina Belmont-Sánchez, Duane Choquesillo-Lazarte, Antonio Frontera, Luis Lezama, Alfonso Castiñeiras, Juan Niclós-Gutiérrez

**Affiliations:** 1Department of Inorganic Chemistry, Faculty of Pharmacy, University of Granada, 18071 Granada, Spain; carol.bs.quimic@hotmail.com; 2Laboratorio de Estudios Cristalográficos (LEC), IACT, CSIC-Universidad de Granada, Av. de las Palmeras 4, 18100 Armilla, Spain; duane.choquesillo@csic.es; 3Department de Química, Universitat de les Illes Balears, Crta. de Valldemossa km 7.5, 07122 Palma de Mallorca, Spain; toni.frontera@uib.es; 4Department of Inorganic Chemistry, Faculty of Science and Technology, University of Basque Country, 48080 Bilbao, Spain; luis.lezama@ehu.es; 5Department of Inorganic Chemistry, Faculty of Pharmacy, University of Santiago de Compostela, 15782 Santiago de Compostela, Spain; alfonso.castineiras@usc.es

**Keywords:** copper(II), linear trinuclear complex, crystal structure, thermogravimetry, spectral properties, magnetism, DFT calculations

## Abstract

Inspired by the reported crystal structure of compound **1**, we aimed to synthesize and determine the structure of compound **2**, where two imidazolium (H_2_im^+^) ions serve as diamagnetic countercations. Here, we report the thermal stabilities, FT–IR, visible, and RSE spectra, as well as the magnetic properties of both compounds. In these structures, µ-EDTA acts as a pentadentate chelator for both terminal Cu centers within the centrosymmetric linear trinuclear anion. The Cu(µ-EDTA) chelates bind to the central Cu(Him)_4_ unit in subtly different ways: in compound **1**, µ-EDTA has a free acetate arm and binds the central Cu(II) center through a syn,anti-carboxylate group. In contrast, in compound **2**, the non-chelating acetate arm serves as a monodentate O-donor to the central Cu(II) atom, increasing the Cu(terminal)···Cu(central) distance from 6.08 Å in **1** to 6.80 Å in **2**. Additionally, pairs of H_2_im^+^ ions in compound **2** display antiparallel π-stacking interactions. We conclude that the H_2_im^+^ counterions in compound **2** enable the magnetic isolation of the nearly identical trinuclear anion present in both compounds. DFT calculations further support the role of different interactions in stabilizing each crystal structure. In compound **2**, dominant contributions from N–H···O hydrogen bonds and π-stacking interactions are accompanied by other, less conventional interactions, such as multiple C–H···O contacts and an O···CO(π-hole) interaction within the trinuclear anion.

## 1. Introduction

Copper is undoubtedly a significant first-row transition element, with relevance in both chemical [1] and biological [2] contexts. The low oxidation state of copper, Cu(II) ([Ar]3d⁹), is particularly intriguing as the “hole formalism” predicts behavior akin to the easily oxidized Ti(III) ([Ar]3d^1^). The coordination chemistry of Cu(II) is extensive, largely due to the static or dynamic distortions imposed by the Jahn–Teller effect, which leads to unequal metal-donor atom distances. Specifically, the six-coordination of Cu(II) often results in various octahedral distortions, such as configurations with four shorter bonds plus two elongated bonds, giving either a symmetric (4 + 2) or asymmetric (4 + 1 + 1) coordination geometry. From this latter arrangement, the loss of the more distant donor atom results in distorted mono-capped square-planar or bipyramidal complexes, while the further loss of a donor produces distorted square-planar or tetrahedral complexes. These possibilities contribute to the diverse physical properties observed in Cu(II) compounds.

In this context, mononuclear to polymeric structures can form, as well as complexes with limited nuclearity (di-, tri-, tetra-, and hexanuclear Cu(II) complexes are quite common). This study focuses on trinuclear Cu(II) complexes with well-characterized crystal structures (see selected examples in references [3,4,5,6,7,8,9,10,11,12,13,14,15,16,17,18,19,20,21,22]). The literature encompasses a variety of linear [3,4,5,6,7,8,9,10,11], angular [12,13,14], or cyclic [14,15,16,17,18,19,20,21,22] trinuclear Cu(II) complexes and highlights three essential aspects: (1) All studies incorporate crystal structure determinations. Among the cited works, one crystallographic study revealed a unique single crystal containing equimolar amounts of both mononuclear and cyclic trinuclear Cu(II) molecules [21]. (2) These compounds hold significant magnetic interest. The essential properties of transition metal compounds enhance their versatility in oxidation states, coordination chemistry, visible coloration, and paramagnetic behavior, making them a focus of extensive research. (3) Theoretical DFT calculations have increasingly been employed to elucidate the role of non-conventional interactions in both molecular and crystal structures. In this context, many of the cited works [3,4,5,6,7,8,9,10,11,12,13,14,15,16,17,18,19,20,21,22] emphasize the importance of correlating crystal structure, magnetic behavior, and DFT calculations to gain a deeper understanding of these systems. An exceptional example of the significance of magnetic studies is the 1987 Nobel Prize awarded to Johannes Georg Bednorz and Karl Alexander Müller for their groundbreaking work on superconductivity—a phenomenon known since 1911 but not fully understood. Their studies on a mixed-valence copper-oxide perovskite (with a crystal structure reported by Bernard Raveau in 1981) revealed a critical temperature of 36 K, below the boiling point of liquid nitrogen (77 K). Currently, significant efforts are being directed toward developing solid phases that exhibit superconductivity at higher temperatures.

The literature on imidazolium (H_2_im^+^) and other protonated amines as countercations for metal complexes is extensive. Notably, a recent study [22,23] reports a giant organic-inorganic hybrid containing H_2_im^+^ cations and [Cu(Br)_4_]^2−^ anions, which exhibit a synergistic motion that affects the shape and size of the single crystal.

In line with our interest in mixed-ligand Cu(II) complexes containing imidazole (Him) and EDTA as co-ligands, we previously reported that the crystal of {[Cu_2_(µ_4_-EDTA)(Him)_2_]·2H_2_O}_n_ [23] exhibits hydrogen bonding, π-stacking, and an unusual (water)O–H···π interaction. Our attention is currently drawn to the work by V. S. Sergienko et al. [6], which includes the synthesis and crystal structure of [Cu(Him)_6_]{μ-Cu(Him)_4_[Cu(EDTA)(Him)]_2_}·6H_2_O (also known as GEMPOE in the Cambridge Structural Database (CSD) and referred to as compound **1** in this study). A search in the CSD (version 5.45, updated June 2024) for closely related transition metal compounds with the same µ-EDTA chelating and bridging role and N-monodentate Him ligand yielded only GEMPOE (compound **1**). A broader search, allowing more flexibility in the chelating and bridging role of EDTA, revealed the crystal structures of trans-[Cu(en)_2_(H_2_O)_2_]{μ-Cu(en)_2_[Cu(µ-EDTA)]_2_}·6H_2_O (BAYLIW10) and the heterometallic trans-[Cu(en)_2_(H_2_O)_2_]{μ-Cu(en)_2_[Ni(µ-EDTA)]_2_}·6H_2_O (AENNIC10). In both cases, EDTA acts as a hexadentate chelator for Cu(II) or Ni(II), contrasting with the pentadentate coordination of EDTA observed in GEMPOE (compound **1**), which also features two monodentate Him ligands instead of the bidentate ethylenediamine (en) used in BAYLIW10 and AENNIC10.

## 2. Results and Discussion

### 2.1. Strategy for the Synthesis of Compound ***1*** and ***2***, Using Basic Copper(II) Carbonates

Sergienko et al. [6] prepared and structurally characterized compound **1** using {[Cu_2_(µ_4_-EDTA)(H_2_O)_2_]·2H_2_O} (known as CUEDTA10 in the CSD [24], later re-determined as CUEDTA01) as a precursor. This compound was suspended in water, and small portions of imidazole were added until a solution formed, which was then left to crystallize at 5 °C. The crude product was recrystallized from water at the same temperature, yielding large crystals, although no yield was reported. The Cu/EDA/Him ratio used (4/2/12.7) represents a ~6% excess of Him compared to the required Cu/EDTA/Him ratio (4/2/12) indicated by the formula of compound **1**. In our opinion, this slight excess does not pose a serious issue, especially given that recrystallization was subsequently performed (see below).

In this work, we prepared both compounds using a similar strategy to those described in recent studies [23,25]. This involved reacting a basic copper(II) carbonate, assumed to be Cu_2_CO_3_(OH)_2_, with the acid form of a chelating amino-polycarboxylic acid (in this case H_4_EDTA, but other acids such as carboxylic acids, amino acids, or oligopeptides can also be used) in water, followed by the addition of the desired N-co-ligand, in this case Him (although many other alkyl or aromatic amines could be used). Two types of copper(II) hydroxy-carbonate are available, which react differently with organic acids: the bluish Cu_2_CO_3_(OH)_2_ (resembling azurite) can yield a brown CuO residue if the reaction is incomplete, while the greenish Cu_2_CO_3_(OH)_2_ (resembling malachite, and now readily available from accredited suppliers) will leave a malachite residue (green color) if added in excess or if the reaction is incomplete, without forming CuO.

The primary advantage of reacting basic copper(II) carbonate with an appropriate amount of organic acid in water is that CO_2_ is the only by-product aside from water, and it can be easily removed. This results in a binary copper(II) complex that can subsequently react with a stoichiometric amount of the desired co-ligand (Him, in the case of both compounds discussed here). The absence of additional by-products (such as inorganic salts) facilitates successful crystallization and increases the likelihood of obtaining suitable single crystals for crystallographic analysis.

For this study, an important consideration was determining the appropriate amount of co-ligand (Him) to use. This was straightforward given the known ratio (Cu/EDTA/Him = 4/2/12) for compound **1** [6] and the corresponding ratio (Cu/EDTA/Him = 6/4/16) for compound **2**, under the assumption that a trinuclear anion [Cu_3_(EDTA)_2_(Him)_2_]^2–^ would crystallize with two H_2_im^+^ cations. This approach presents the intriguing possibility of using the protonated form of the N-ligand (Him, taking part in the trinuclear anion) as its countercation (H_2_im^+^).

Assuming ‘two times’ the simplest reaction CO_3_^2−^ + 2 OH^−^ + 4 H^+^ → CO_2_ ↑ + 3 H_2_O, the stoichiometric process that produces the reported compound **1** can be expressed as follows:2 Cu_2_CO_3_(OH)_2_ + 2 H_4_EDTA + 12 Him → 
            [Cu(Him)_6_]{Cu(Him)_4_[Cu(µ-EDTA)(Him)]_2_}·6H_2_O (**1**) + 2 CO_2_ ↑

Similarly, for “three times” the reaction CO_3_^2−^ + 2 OH^−^ + 4 H^+^ → CO_2_ ↑ + 3 H_2_O and considering the resulting ratio Cu/EDTA/Him = 6/4/16, the overall stoichiometric process that yields the novel compound **2** is as follows:3 Cu_2_CO_3_(OH)_2_ + 4 H_4_EDTA + 16 Him → 
              2 (H_2_im)_2_{Cu(Him)_4_[Cu(µ-EDTA)(Him)]_2_}·2H_2_O (**1**) + 3 CO_2_↑ + 2 H_2_O

Of the 16 H^+^ ions from 4 H_4_EDTA, four are used to form four imidazolium(1+) cations, while the remaining twelve generate six water molecules. Of these, four crystallize with compound **2**, and two constitute a negligible part of the solvent.

Our results showed that, as reported by Sergienko et al. [6], compound **1** produced large crystals that were soluble in water, with solubility increasing appreciably with a slight rise in ambient temperature. This can be seen in Figure 1a,b, where polycrystalline masses coexist with elongated crystals of compound **1**, along with a photograph of the single crystal used to confirm the identity of compound **1**, as previously reported. In contrast, Figure 1c shows an abundance of well-formed crystals of compound **2**, which were stable in air at room temperature, allowing for their crystallographic study to be conducted at room temperature.

### 2.2. Crystal Structures of ***1*** and ***2***

#### 2.2.1. Crystal Structure of Compound **1**

Since the crystal structure of compound **1** has been reported [6] and is available in the CSD database as GEMPOE, in Appendix A, we report additional information. This compound (measured at 295 K) crystallizes in the monoclinic system, space group *P*$\bar{1}$ and its structure was refined to an *R*_1_ = 0.047, compared to those of the crystal of compound **2** (*R*_1_ = 0.043, see below). Figure 2 shows the ions present in this crystal (along with six water molecules not bound to the metal, omitted here for clarity).

It is interesting to take into account that all Cu(II) centers exhibit elongated octahedral coordination, ~type 4 + 2, being four coordination bonds of ~2 Å nearly coplanar and the remaining two (so-called distal) of ~2.38–2.62 Å. The Cu1 atom of cation the [Cu(Him)_6_]^2+^ (trans-distal bond Cu1-N1 of 2.469(5) Å) and the Cu2 central atom of the linear trinuclear anion (trans-distal bonds Cu2-O6 of 2. 615(5) Å) have centrosymmetric environments, whereas the coordination of the Cu3 terminal atoms has slightly unequal trans-distal bonds: Cu3-O1(EDTA) 2.385(5) Å and Cu-N9(EDTA) 2.397(5) Å.

In **1**, the linear trinuclear anion is centrosymmetric, and the EDTA acts as a pentadentate chelator to the terminal Cu3 centers (which accomplishes their six-coordination with a Cu3-N7(Him) bond) and also monodentate, by the bond O6-Cu2(central) of 2.615(5) Å. This implies that the link between the terminal Cu3 center and the central Cu2 atom is accomplished by the anti,syn-bridging carboxylate group, remaining EDTA with a free acetate arm All that ensures the linearity of the trinuclear Cu(II) anion, with two equal Cu2···Cu3 inter-nuclear distances of 6.084(3) Å, whereas the two shortest Cu1(cation)···Cu3(terminal anion) distances are of 9.724(4) or 9.840(3) Å. In this crystal, each trinuclear anion is H-bonded to six surrounding cations, and each cation is H-bonded to four neighboring anions. Both kinds of ion complexes and non-metal bonded water molecules construct the three-dimensional (3D) crystal packing as dominant weak interactions. Additional structural information is provided in Appendix A.

#### 2.2.2. Crystal Structure of Compound **2**

The crystal structure of this compound reveals a centrosymmetric, linear trinuclear anion accompanied by two imidazolium(1+) cations and two disordered water molecules (Figure 3). Table 1 summarizes the relevant distances and angles involving non-hydrogen atoms. All metal centers exhibit an elongated octahedral coordination of the 4 + 2 type. Each of the two terminal Cu1 centers is coordinated by an N-donor from an imidazole (Him) group and five other donors from the EDTA ligand (two N-amino and three O-carboxylate). Notably, the remaining acetate arm of the EDTA ligand in the terminal moieties acts as a monodentate O-donor to the central Cu2 atom, occupying its two trans-distal coordination sites with relatively long bonds, specifically Cu2–O19 at 2.720(3) Å. Due to this μ-EDTA coordination mode, the intermetallic Cu1···Cu2 distance in this trinuclear anion (6.803(1) Å) is significantly larger than that in compound **1** (6.084(3) Å), where the μ-EDTA ligand links each terminal Cu center to the central Cu via an anti,syn-carboxylate bridging group. In the novel compound **2** (Figure 4), the symmetrical coordination around the central Cu2 atom is achieved through four N-Him donors, forming two pairs of nearly coplanar bonds (Cu2–N30 at 1.996(2) Å and Cu2–N25 at 2.033(2) Å), as detailed in Table 1.

In the crystal packing of **2**, the π-stacking between H_2_im^+^ cations (Figure 5) works in tandem with numerous conventional hydrogen bonds (Figure 6) and other weaker interactions. The β and γ angles, as defined in Figure 5a, are both equal to 3.81° (see Appendix A), confirming the face-to-face nature of this π-stacking interaction. The disorder of the water molecules is due to the implication of the H atoms water (disordered as O1A and O1B) as donor to non-bonded to Cu carboxylate atoms O9 and O20#6 (#6 = −x, −y + 1, −z + 1) as acceptors. A number of C-H···O interactions can be considered (Appendix A), moreover complementary structural information is provided in Appendix A.

#### 2.2.3. Structural Impact of Using Imidazolium(1+) Ions to Achieve Magnetic Isolation of the Linear Trinuclear Anion in Compound **1**

The crystal structures of the two compounds under study exhibit both similarities and differences that are crucial to understanding their physical properties and interpreting the results of DFT calculations.

Among the similarities, we note the following: (1) All Cu(II) centers display a somewhat elongated octahedral coordination geometry (approximately 4 + 2 type). (2) Both compounds contain linear trinuclear Cu(II) complex anions. (3) The EDTA acts as a pentadentate chelator to the terminal Cu atoms of the anions, which are also centrosymmetric. (4) Conformational analysis of the chelation mode revealed a stable conformation, primarily determined by the puckered Cu(N-N) E ring, which is coplanar with a G Cu(glycinate) ring, leaving two Cu(glycinate) R rings perpendicular to both the E and G rings (see Appendix A). This arrangement allows a Him ligand to bind to one of the four closest sites on the terminal Cu center. (5) The central Cu centers have four Him ligands occupying the four closest and coplanar coordination sites. The preference of the Him ligand for the closest coordination to Cu(II) centers is easily explained by Pearson’s Hard and Soft Acids and Bases (HSAB) theory, as Cu(II) is a borderline acid and the N-heterocyclic donor of Him is a well-known borderline base. Additionally, the Jahn–Teller effect causes a 4 + 2 distortion in the [Cu(Him)_6_]^2^^+^ ion.

The primary differences between the crystal structures of compounds **1** and **2** are as follows: (a) The bridging role of μ-EDTA in connecting Cu(terminal) and Cu(central) centers differs; in compound **1**, it links via an anti,syn-carboxylate group (with a free acetate arm), whereas in compound **2**, it uses a monodentate acetate arm (not involved in chelation). This change increases the Cu(terminal)···Cu(central) separation from 6.084(1) Å in compound **1** to 6.803(1) Å in compound **2**. (b) A key distinction of this study is the paramagnetism of cation in compound **1** versus the diamagnetism of the H_2_im^+^ cation in compound **2**. (c) The compounds exhibit different amounts of unbound water molecules. (d) π-stacking interactions between pairs of H_2_im^+^ ions contribute to the crystal packing in compound **2**, a feature not observed in compound **1**.

### 2.3. Physical Properties

#### 2.3.1. Thermogravimetric Studies, with Identification of Evolved Gases and Estimation of the CuO as Final Residues

Figure 7 presents the weight loss curves from the thermogravimetric analyses (TGAs) of compounds **1** (left) and **2** (right), with vertical lines marking the start and end of the stages identified based on their corresponding first derivatives Table 2 and Table 3 summarize the data obtained from these analyses. Additional details illustrating the elaboration of these Tables are provided in Appendix A, with a more detailed version for compound **2**.

The TGAs highlight the difficulty in pinpointing the end temperature for the release of non-coordinating water molecules (six for **1** and two for **2**), which would typically be expected below 100 °C. However, this stage extends up to approximately 180 °C, with some CO_2_ released during this phase. Notably, discrepancies between experimental and theoretical weight loss values exceed the typical margin of ~1%. The final experimental residue, identified as CuO, matches the calculated value with less than 1% deviation, with the experimental result being slightly higher.

#### 2.3.2. Infrared (FT–IR) Spectra of Polycrystalline Samples

The spectra, including wavenumber readings (cm^−1^), are provided as Appendix A. Key diagnostic bands referenced during the synthesis methods (Section 3.1.1 and Section 3.1.2 for compounds **1** and **2**, respectively) are highlighted. The complexity of both spectra is evident, and, as in previous work by our group [20,23,25], they serve the dual purpose that infrared spectroscopy is well known for. Firstly, during crystallization, small samples (a few mg) are periodically taken using a Pasteur pipette, and their FT–IR spectra are recorded. Identical spectra confirm that the samples correspond to the same compound. Secondly, characteristic bands with high diagnostic values are identified.

Two notable points emphasize the importance of this analysis. First, it is well established that in carboxylates or EDTA chelates (or other amino-carboxylates, see Appendix A), the difference between the antisymmetric and symmetric frequencies (Δ(ν_as_ − ν_s_)) of a carboxylate group varies depending on its coordination type—monodentate, bridging (anti,syn or anti,anti- or syn,syn-), O,O′-chelating to the same metal center, or ionic (uncoordinated). For compounds containing the same metal ion, such as copper(II) here, Δ(ν_as_ − ν_s_) decreases in the order: monodentate > bridging > O,O′-chelating > ionic. In compound **1**, carboxylate groups of three types are present (anti,syn-bidentate, monodentate, and free), whereas, in compound **2**, all carboxylate groups function as monodentate, either for terminal or central Cu centers. Second, compound **2** contains both imidazole and imidazolium(1+), whose vibrational behavior is often underexplored in many references. The >N + H chromophore produces a series of weak but distinct peaks in the ~2300–2700 cm^−1^ region, visible in the spectrum of compound **2** (see Appendix A) and absent in compound 1. This difference is noteworthy, as the spectrum of compound **1** is from a more concentrated sample (see Appendix A).

#### 2.3.3. Electronic (Diffuse Reflectance) Spectra

Figure 8 presents the electronic spectra of compounds **1** and **2**. Both compounds display a broad and intense band around 700 nm, accompanied by a wider absorption with a maximum of approximately 1100 nm. This spectral behavior is similar to that observed for the [Cu(H_2_O)_6_]*^2+^* ion in solution, which shows maxima at around 800 and 1060 nm [1]. Such features are typical for Cu(II) complexes with elongated octahedral coordination (type ~4 + 2).

#### 2.3.4. Electron Spin Resonance (ESR) Spectra and Magnetic Properties

The X and Q-band ESR spectra registered at room temperature of compound **1** (GEMPOE) are displayed in Figure 9. The spectra are rather complex due to the superposition of signals corresponding to different magnetic centers, but in the low-field region, the typical four-line hyperfine splitting originated from the interaction of the unpaired electron on Cu^2+^ with the ^63^Cu and ^65^Cu nuclei (I = 3/2) is unequivocally identified. This fact indicates that one of the Cu^2+^ ions of the complex is magnetically isolated from the rest, and considering the structure of this compound, it can only be the one belonging to the [Cu(Him)_6_]^2+^ cation. Moreover, the hyperfine structure of the spectra can be simulated with the following spin Hamiltonian parameters: g_II_ = 2.301; g_⊥_ = 2.085; A_II_ = 166 × 10^−4^ cm^−1^, in good agreement with the elongated octahedron formed by six N atoms around the cooper atom in this monomer [26]. The additional shoulders detected on the Q band spectra, at about 11,000 and 11,350 Gauss, belong to the signal generated by the trimeric fragment of the compound. Therefore, there is no operational exchange pathway between the anionic trinuclear part of the compound and the cationic unit.

The X-band room temperature spectrum of **2** shows only one apparently quasi-axial signal, but operating at the Q-band, the superposition of two different signals can be clearly observed (Figure 10). By computer simulation of this spectrum, the following main components of the g-tensors were determined:(1) g_1_ = 2.302; g_2_ = 2.105; g_3_ = 2.051; (2) g_1_ = 2.235; g_2_ = 2.082; g_3_ = 2.081

These values agree well with those expected, considering the characteristics of the coordination polyhedra around Cu1 and Cu2 atoms in this complex, respectively [27]. The absence of evidence for hyperfine structure and/or ΔMs = ±2 signals indicates that the magnetic exchange is not negligible, although it is very weak. The difference between the X- and Q-band spectra implies that the exchange interaction is strong enough to average the individual signals operating at the X-band but not at the Q-band [8]. Therefore, the magnitude of the exchange interaction between the Cu(II) ions in the trinuclear unit of **2** has to be about 0.1 cm^−1^.

The magnetic behavior of compounds **1** and **2** is shown in Figure 11 in the form of Χ*_m_*^−1^ and Χ*_m_T* versus *T* plots, where Χ*_m_* is the magnetic molar susceptibility. In both cases, the susceptibility data are well described by means of the Curie–Weiss expression in practically all the recorded temperature ranges, with C_m_ = 1.30 cm^3^Kmol^−1^; θ = +0.1 K for **1** and C_m_ = 1.69 cm^3^Kmol^−1^; θ = −0.1 K for **2**. The C_m_ values are in good agreement with that expected for four and three magnetically non-interacting copper(II) ions, respectively. The magnetic effective moments remain practically constants down to 2 K, and they only seem to increase (**1**) or decrease (**2**) very slightly at the lowest temperatures. This behavior is in agreement with what was observed in the ESR spectra. Despite the relatively short Cu···Cu distances within the trimers (6.083(1) Å for **1** and 6.8031(1) Å for **2**), the magnetic interactions are very weak due to unfavorable exchange pathways. The couplings have to propagate through the d_z2_ orbitals, with large bond distances (2.720(3) and 2.61 Å for **1** and **2**, respectively), while the unpaired electrons occupy mainly d_x2–y2_ orbitals [28].

### 2.4. Theoretical DFT Studies

The theoretical study is focused on the analysis of several supramolecular assemblies observed in the solid state of compounds **1** and **2**. In particular, we are interested in the characterization of the assemblies shown in Figure 12, where the main difference between compounds **1** and **2** regarding the trinuclear complex is that in **2**, the anionic part forms self-assembled dimers directed by strong NH···O H-bonds and also O···C interactions where the O-atom of one carboxylate group points to the carbon atom of the carboxylate group of the adjacent molecule, thus establishing interesting O···π-hole interactions. In contrast, compound **1** does not form such assemblies, and the trinuclear anionic part interacts with the cationic one. Moreover, compound **2** also forms supramolecular polymers (see Figure 12c) that are propagated by a combination of H-bonds and π–π stacking interactions.

Figure 13 shows the MEP surfaces of a model of compound **2**. Such a model is convenient for simplicity and uses a neutral fragment. That is, the trinuclear complex is a dianion, thus not adequate to compute the MEP and adequately differentiate the electron-rich and poor parts of the molecule. The model used is depicted in Figure 10, top-right. The MEP maximum is located at the imidazolium NH (+65.9 kcal/mol) and the minimum at the O-atoms of the carboxylate groups, as expected (−62.7 kcal/mol). Remarkably, the MEP is also large and positive at the NH of the coordinated imidazole ring (+63.9 kcal/mol), likely due to its coordination with the Cu(II) metal center. The MEP values over the coordinated imidazole ring are slightly negative and positive over the protonated imidazolium (−6.2 and +26.4 kcal/mol, respectively).

First, we have analyzed the π-stacking/H-bonding assembly of compound **2** (see Figure 14) by using the model depicted in Figure 13 (top-right) to keep the size of the system computationally approachable. To characterize the interactions, the quantum theory of atoms in molecules (QTAIM) and the noncovalent interaction plot (NCI plot) computational tools were combined because they are convenient for revealing NCIs in real space. In this type of plot, blue and green colors are used for strong and weak interactions, respectively. Figure 14 shows the QTAIM analysis of bond critical points (CPs, represented as small red spheres) and bond paths (represented as orange lines). It can be observed that the imidazolium cation is connected to the Cu-complex by means of two bond CPs and bond paths connecting both O-atoms of the carboxylate groups to the NH and CH groups of the cation, thus revealing that in addition to the strong NH···O, a weaker CH···O is also formed. This is confirmed by the reduced density gradient (RDG) isosurfaces that are located coincident with the bond CPs. They are blue for the NH···O contact and green for the CH···O one. The combined QTAIM/NCIplot also confirms the existence and attractive nature of the π-stacking. The shape of the RDG isosurface reveals the large overlap of the π-systems. DFT calculations combined with the QTAIM potential energy densities disclose that the NH···O is the strongest interaction (−10.7 kcal/mol) followed by the π-stacking (−4.5 kcal/mol) and the CH···O (−1.1 kcal/mol).

Figure 15 shows the QTAIM/NCIplot representation of two selected assemblies of compounds **1** and **2**. For compound **2**, we have selected the self-assembled dimer of the trinuclear fragment in order to further corroborate the existence of the O···C π-hole interactions. As can be observed in Figure 15a, there is an intricate network of contacts connecting both fragments (up to 12 bond CPs and bond paths). The NH···O interactions are highlighted in yellow, showing that each H-bond is characterized by one bond CP, bond path, and bluish isosurface connecting the O and H-atoms. The O···CO contacts are highlighted in pink, evidencing that each π-hole interaction is characterized by a bond CP and bond path interconnecting the C and O-atoms. Moreover, the green color of the RDG isosurface confirms its attractive nature. The QTAIM/NCIplot analysis reveals the existence of several C–H···O H-bonding interactions that are characterized by the corresponding bond CPs, bond paths, and green RDG surfaces interconnecting the H and O-atoms. Finally, two bond CPs and bond paths also interconnect two coordinated imidazole rings, disclosing the presence of a parallel-displaced π-stacking interaction. The energies associated with each interaction are also included in Figure 15a, evidencing that the NH···O are the strongest contacts, followed by the contribution of the six CH···O interactions (−8.6 kcal/mol in total). Both the π-stacking and π-hole interactions are weaker, −1.2 kcal/mol and −1.6 kcal/mol, respectively. For compound **1**, a similar study has been performed, showing that the NH···O interaction is the strongest one (−2.2 kcal/mol). The QTAIM/NCIPlot shows that for this assembly, the imidazole···imidazole interactions are abundant, basically C–H···π interactions (characterized by three bond CPs and bond paths), which contribute in −1.7 kcal/mol. Finally, an additional weak C–O···H contact (−0.5 kcal/mol) connects one imidazole ring of the cationic fragment to the trinuclear complex. Although the sum of the energetic contributions is much smaller in **1** than in **2**, the formation of the dimer of **1** is strongly favored electrostatically since the trinuclear Cu-complex is dianionic and the mononuclear Cu complex is dicationic. In contrast, in compound **2**, the formation of the dimer is against the electrostatic repulsion of the dianionic trinuclear Cu-complexes.

## 3. Materials and Methods

### 3.1. Strategy and Procedure for the Synthesis of Both Studied Compounds

The reagents Cu_2_CO_3_(OH)_2_ (malachite, Aldrich, Merck Life Science S.L.U., María de Molina, 40-2 Planta, Madrid 28006, Spain), H_4_EDTA (Merk, Merck Life Science S.L.U., María de Molina, 40-2 Planta, Madrid 28006, Spain), and Him (Merk), and anti-solvents (diethyl ether and isopropanol) were purchased for commercial sources.

The synthesis of both studied compounds was inspired by those reported for related compounds {[Cu_2_(µ_4_-EDTA)(Him)_2_(H_2_O)_2_]·2H_2_O}_n_ [23] and [Cu_2_(µ_2_-EDTA)(9heade)_2_ (H_2_O)_4_]·3H_2_O [25], essentially consisting in a two-step process. First, the reaction of H_4_EDTA and malachite Cu_2_CO_3_(OH)_2_ (the only basic carbonate of copper(II) with appropriate quality now commercially available) in water, with heating (~50 °C) and magnetic stirring, which produces CO_2_ as a by-product. And secondly, the addition of the required stoichiometric amount of Him causes an intensification of the blue color. If the first step is complete, the slow filtration of the resulting solution does not leave a perceptible green residue in the filter Kitasato funnel. If that is carried out, the solution is collected in an Erlenmeyer flask, where Him is added. After the reaction, the mother liquors were collected in an appropriate crystallizer flask, where solid (crystalline?) samples would be formed. Successive samples were checked by FT–IR to see whether the samples corresponded to the same phase. In the case of compound **2**, the excellent crystallization also enables us to check that only a crystalline phase is formed. Hence, the use of single crystals with the employed diffractometer (see below, in Section 3.2) easily confirms that they correspond to compound **2**.

#### 3.1.1. Synthesis of [Cu(Him)_6_]{μ-Cu(Him)_4_[Cu(EDTA)(Him)]_2_}·6H_2_O (**1**), with Relevant Vis-UV and FTIR Spectral Data

Malachite Cu_2_CO_3_(OH)_2_ (0.44 g, 2 mmol) and H_4_EDTA (0.59 g, 2 mmol) were reacted in 75 mL of water until no insoluble green residue was perceptible. Heating and stirring are continued for a half hour. Then, Him is added (0.82 g, 12 mmol), and the mixture is stirred for a few minutes until a clear blue solution forms. The heating is interrupted, and the mother liquor is filtrated again on a crystallization flask, which is covered with a plastic film, in such a manner that enables a slow evaporation of the solvent. Note that this reaction mixture has the stoichiometric ratio Cu/EDTA/Him 4/2/12. When the initial volume is reduced to three-thirds, the flask is placed into a desiccator to slowly diffuse vapors of a mixture of diethyl ether and isopropanol, which induces the crystallization of the desired product (**1**). The performance in repeated experiences exceeded 65%. Caution! However, if the crystallizing device is left (covered by tight plastic film) at room temperature, thermal fluctuations re-dissolve and again recrystallize the product, noticeably altering its crystallographic quality. Elemental analysis (%): Calc. for C_56_H_84_Cu_4_N_28_O_22_: C 38.31, H 4.82, N 22.34, Cu (as CuO) 18.12; Found: C 38.27, H 4.79, N 22.28, Cu 18.34 (as CuO, final residue at 670 °C, in its TGA curve). FT–IR data (cm^−1^): For H_2_O: ~3650–~2500. For H_2_O ν_as_ 3372, ν_s_~2300, δ 1603; for Him ν(N-H), 3142, 3126, 3120,ν(C-H), 3043, aromatic skeletal peaks 1823, 1777; δ(N-H) 1541w, 1507w; π(C-H) 860 (one C-H, expected at 900–860) and 836 (two adjacent C-H, expected at 860–810); for EDTA ν_as_(CH_2_), 2934, ν_s_(CH_2_), 2857, 2805; carboxylate groups: anti,syn-bridge-COO, ν_as_(COO) 1644, ν_s_(COO) 1397, Δ(COO) 247; chelating monodentate-COO ν_as_(COO) 1582, ν_s_(COO) 1387, Δ(COO) 195, free-COO ν_as_(COO) 1582, ν_s_(COO) 1372, Δ(COO) 210. Electronic (diffuse reflectance) spectrum: Band with a maximum at 710 nm and accompanied by a shoulder ~1064 nm.

#### 3.1.2. Synthesis of Compound (H_2_im)_2_[Cu(Him)_4_{(µ-EDTA)Cu(Him)}_2_]·2H_2_O (**2**), with Relevant Vis–UV and FT–IR Spectral Data

This product was obtained in a manner similar to compound **1**, using Cu_2_CO_3_(OH)_2_ (0.3 g, 1.5 mmol), H_4_EDTA (0.59 g, 2 mmol), and Him (0.55 g, 8 mmol) in water 75 mL; that is in a Cu/EDTA/Him ratio 3/2/8. In spite of an excellent crystallization, the product is also extremely soluble in water and was collected with yields of 40–70%. Elemental analysis (%): Calc. for C_44_H_62_Cu_3_N_20_O_18_: C 39.15, H 4.63, N 20.75, Cu (as CuO) 17.68; Found: C 39.10, H 4.46, N 20.68, Cu 17.87 (as CuO, final residue at 670 °C, in its TGA curve).

FT–IR data (cm^−1^): a very broad absorption at ~3650~2500. For H_2_O: ν_as_~3418, ν_s_~2250, δ~1600; for H_2_im^+^ and Him: ν(N-H), 3141, 3115, ν(C-H), 3059, 3043; only for N^+^H a series of defined and weak peaks between 2750 and 2500!, δ(N-H) 1543, 1506; for only one aromatic C-H, π(C-H) 877, 859 (expected at 900–860), 899, 877; for two adjacent aromatic C-H, π(C-H) 837, 817 (expected at 860–810); for EDTA ν_as_(CH_2_), 2956, 2925, ν_s_(CH_2_), 2846, 2802; carboxylate groups: chelating monodentate-COO ν_as_(COO) 1602, ν_s_(COO) 1385, Δ(COO) 219, free-COO ν_as_(COO) 1543, ν_s_(COO) 1385, Δ(COO) 158. Electronic (diffuse reflectance) spectrum: Band with a maximum at 690 nm, accompanied by a shoulder ~1100 nm.

### 3.2. Crystal Structure Determination of (H_2_im)_2_[Cu(Him)_4_{(µ-EDTA)Cu(Him)}_2_]·2H_2_O *(**2**)*

As the crystal structure of compound Cu(Him)_6_] {μ-Cu(Him)_4_[Cu(EDTA)(Him)]_2_}·6H_2_O (**1**, GEMPOE in CSD) has been previously reported by Sergienko et al. [6]. Appendix A provides their crystal data, experimental details and structure refinement, coordination bond lengths and angles, and H-bonding interaction obtained with its deposited cif in CSD.

A blue prismatic crystal of (H_2_im)_2_[Cu(Him)_4_{(μ-EDTA)Cu(Him)}_2_]·2H_2_O was mounted on a glass fiber and used for data collection. Crystal data were collected at 298(2) K, using a Bruker (Bruker AXS, Östliche Rheinbrückenstr. 49, 76187 Karlsruhe, Germany) D8 VENTURE PHOTON III-14 diffractometer. Graphite mono-chromate [CuK(α)] radiation (λ = 1.54184 Å) was used throughout. The data were processed with APEX3 [29] and corrected for absorption using SADABS (transmissions factors: 1.000–0.890) [30]. The structure was solved by direct methods using the program SHELXS-2013 [31] and refined by full-matrix least-squares techniques against *F*^2^ using SHELXL-2013 [31]. Positional and anisotropic atomic displacement parameters were refined for all non-hydrogen atoms. Hydrogen atoms were located in difference maps and included as fixed contributions riding on attached atoms with isotropic thermal parameters 1.2/1.5 times those of their carrier atoms. The criteria for a satisfactory complete analysis were that the ratios of root mean squares shift to standard deviation less than 0.001, and there are no significant features in the final difference maps. Atomic scattering factors were taken from the International Tables for Crystallography [32]. Molecular graphics were plotted with DIAMOND [33]. Crystal data, experimental details, structure refinement, coordination bond lengths and angles, and H-bonding interactions are provided (Appendix A). Crystallographic data for compound **2** have been deposited in the Cambridge Crystallographic Data Centre, with the reference number 2400561. Copies of this information could be obtained free of charge upon application at http://www.ccdc.cam.ac.uk/products/csd/request (accessed on 17 March 2024).

### 3.3. Other Physical Measurements

The elemental analysis was performed with a Thermo Scientific Flash 2000 (Thermo Fisher Scientific Inc., Waltham, MA, USA). Infrared spectra (samples in KBr pellets) were recorded using a Jasco FT-IR 6300 spectrometer (Jasco Analítica, Madrid, Spain). Electronic (diffuse reflectance) spectra were obtained in a Varian Cary-5E spectrophotometer (Agilent Scientific Instruments, Santa Clara, CA, USA) from ground crystalline samples. Thermogravimetric analyses (TGA) were carried out (10 °C/min) under air-dry flow (100 mL/min) with a thermobalance Mettler-Toledo TGA/DSC1 (Mettler-Toledo, Columbus, OH, USA), and a series of 35 time-spaced FT–IR spectra were recorded to identify evolved gasses throughout each experiment, using a coupled FT-IR Nicolet 550 spectrometer (Thermo Fisher Scientific Inc., Waltham, MA, USA).

### 3.4. ESR Spectra and Magnetism

X-band ESR spectra of powdered samples were recorded on a Bruker ELEXSYS E500 spectrometer equipped with a super-high-Q resonator ER-4123-SHQ. The magnetic field was calibrated by an NMR probe, and the frequency inside the cavity was determined using an integrated MW-frequency counter. Q-band measurements were carried out by a Bruker ESP300 spectrometer equipped with an ER-510-QT resonator, a Bruker BNM 200 gaussmeter, and a Hewlett-Packard 5352B microwave frequency counter. Data were collected and processed using the Bruker Xepr suite. Magnetic measurements of powdered samples were performed in the 2–300 K temperature range by using a Quantum Design MPMS3 SQUID magnetometer under a magnetic field of 0.1 T (diamagnetic corrections were estimated from Pascal’s constants). The experimental susceptibilities for the diamagnetism of the constituent atoms were corrected using Pascal tables.

### 3.5. Computational Methods

The calculations using the crystallographic coordinates were carried out using the Turbomole 7.2 program [34]. The level of theory used for these optimizations was PBE0-D3/def2-TZVP [35,36,37]. The MEP surface plots were computed at the same level as the theory, and the 0.001 a.u. isosurface. The analysis of the electron charge density was performed using the quantum theory of atoms in molecules (QTAIM) [38] and complemented by the noncovalent interaction plot index (NCIplot) [39] by using the reduced density gradient (RDG) isosurfaces. They were plotted using the VMD program 1.9.3 [40]. The settings for the RDG plots were: *s* = 0.5 a.u.; cut-off ρ = 0.05 a.u., and color scale −0.04 a.u. ≤ sign(λ_2_)ρ ≤ 0.04 a.u.

## 4. Conclusions

Our efforts to magnetically isolate the linear trinuclear complex [Cu(Him)_6_] {μ-Cu(Him)_4_[Cu(EDTA)(Him)]_2_}·6H_2_O (**1**) in the form of the novel imidazolium(+) salt (H_2_im)_2_[Cu(Him)_4_ {(µ-EDTA)Cu(Him)}_2_]·2H_2_O (**2**) have been reasonably successful. However, the use of distinct countercations results in different crystal packing arrangements, which notably influence the bridging role of the μ-EDTA. This leads to two significant structural features: Firstly, the μ-anti,syn-carboxylate group of μ-EDTA in **1** produces a shorter intermetallic Cu(terminal)···Cu(central) separation compared to the monodentate O-acetate bridge used in **2**. Secondly, the anion-cation recognition in these compounds effectively enables the magnetic isolation of the trinuclear anion in **2** while also impacting the crystal packing, with corresponding effects on physical properties.

Theoretical studies reveal strong H-bonds and π-stacking interactions in compound **2**. Additionally, self-assembled dimers of the anion in **2** are stabilized partly by unconventional O···CO π-hole interactions, as confirmed by QTAIM and NCIplot computational tools. Finally, the computed interactions indicate that NH···O H-bonds are dominant; however, numerous CH···O interactions also play a critical role in the solid-state architecture of **2**.

## Figures and Tables

**Figure 1 ijms-25-13130-f001:**
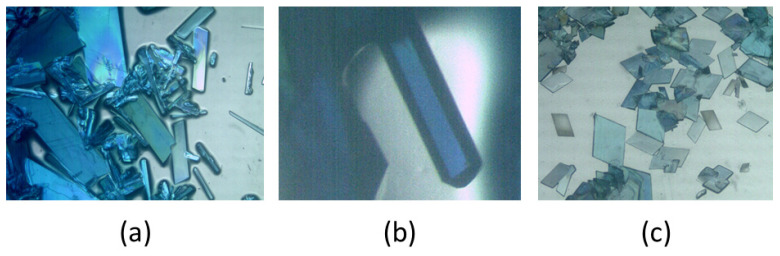
Crystals of compounds **1** (**a**,**b**) and **2** (**c**).

**Figure 2 ijms-25-13130-f002:**
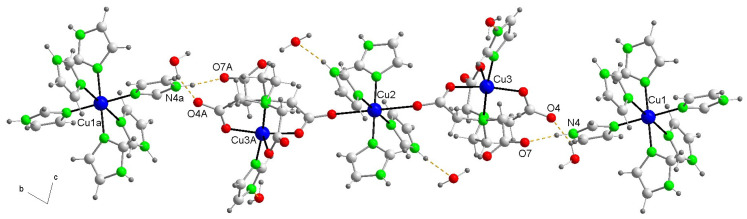
Structure of compound **1** showing two cations linked to a trinuclear anion by H-bonds. The shortest intermetallic distances between Cu centers here are Cu2(central)···Cu3(terminal) within the trinuclear anion 6.084(3) Å and Cu1(cation)···Cu3(terminal anion) 9.724(4) Å. Color code: Cu, blue; C, pale gray; O, red; N, bright green, H, dark gray.

**Figure 3 ijms-25-13130-f003:**
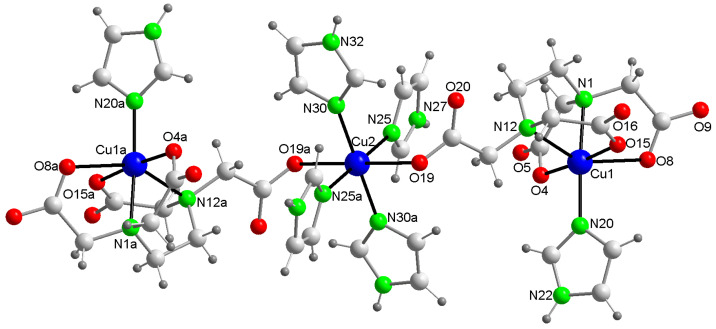
Structure of the linear trinuclear anion in the crystal of compound **2**. Non-coordinated water omitted. Symmetry code #1 = −x, −y + 2, −z. Color code: Cu, blue; C, pale gray; O, red; N, bright green, H, dark gray.

**Figure 4 ijms-25-13130-f004:**
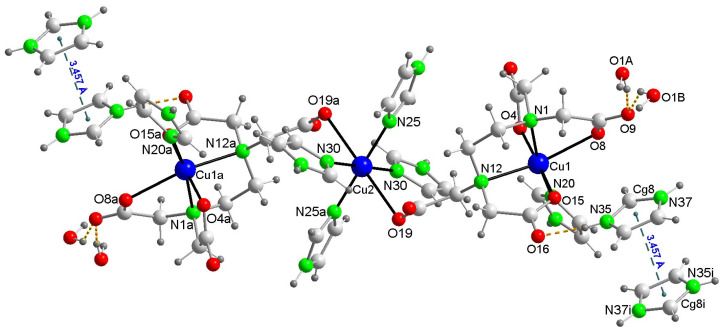
Plot of compound **2** showing the π-stacking between pairs of H_2_im^+^ cations. Symmetry code i = −x, 1 − y, −z. Color code: Cu, blue; C, pale gray; O, red; N, bright green, H, dark gray.

**Figure 5 ijms-25-13130-f005:**
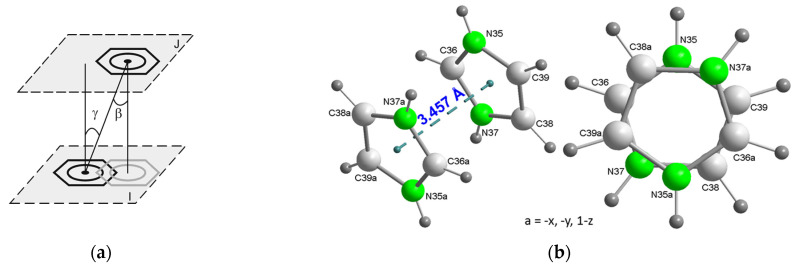
Stacking interaction between pairs of H_2_im^1+^ cations in the crystal of compound **2**. (**a**) Scheme showing the π-stacking parameters, in this case, between two aromatic six-membered rings, both of them being supposedly parallel (and hence defining a dihedral angle α between the panes I and J of 0°). (**b**) Pairs of stacked H_2_im^1+^ counterions in the crystal of **2**, showing (**left**) the inter-centroid distance and (**right**) the antisymmetric orientation of both H_2_im^+^, related by an inversion center. For symmetry reasons, each π-stacked imidazolium ions fall in parallel planes, with α = 0°. Color code: C, pale gray; N, bright green, H, dark gray.

**Figure 6 ijms-25-13130-f006:**
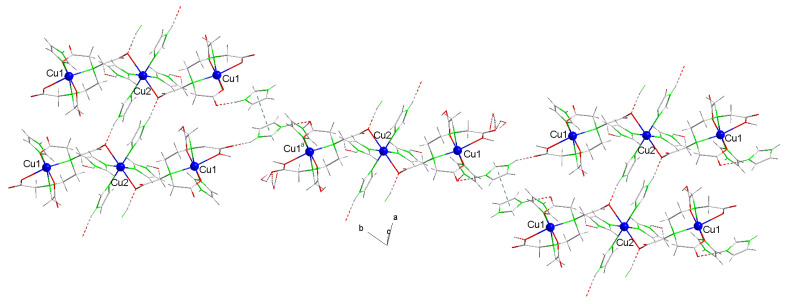
Conventional H-bonds also contribute to the crystal packing of **2**. Each trinuclear anion is surrounded by two pairs of π-stacked imidazolium cations.

**Figure 7 ijms-25-13130-f007:**
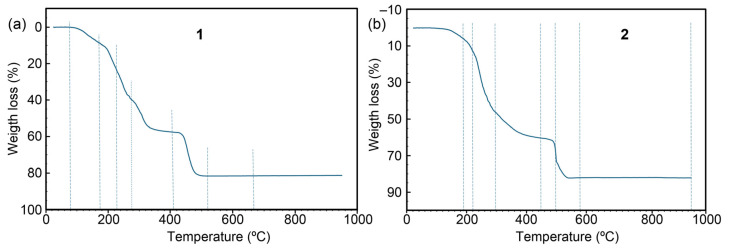
Pattern of the weight loss curves in the thermogravimetric analyses (TGAs) of compounds **1** (**a**) and **2** (**b**). Vertical lines indicate each step of the TGA.

**Figure 8 ijms-25-13130-f008:**
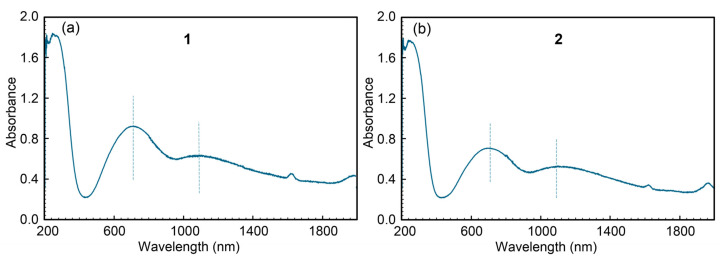
Electronic spectra of compounds **1** (**a**) and **2** (**b**). Vertical lines indicate the location of the maximum of some bands.

**Figure 9 ijms-25-13130-f009:**
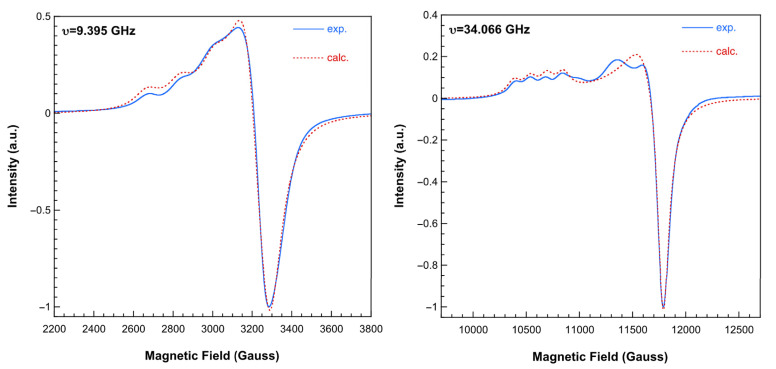
X-band (**left**) and Q-band (**right**) ESR spectra of complex **1** recorded at room temperature. Dashed lines correspond to the calculated spectra for the monomeric fragment of the compound (see text for the details).

**Figure 10 ijms-25-13130-f010:**
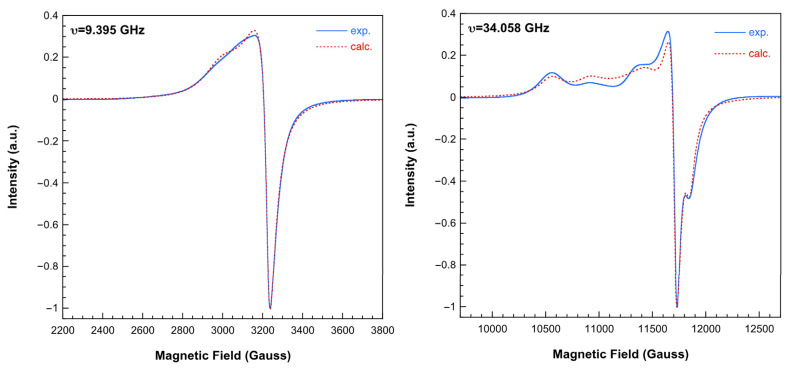
X-band (**left**) and Q-band (**right**) ESR spectra of compound **2** recorded at room temperature. Dashed lines correspond to calculated spectra (see text for the details).

**Figure 11 ijms-25-13130-f011:**
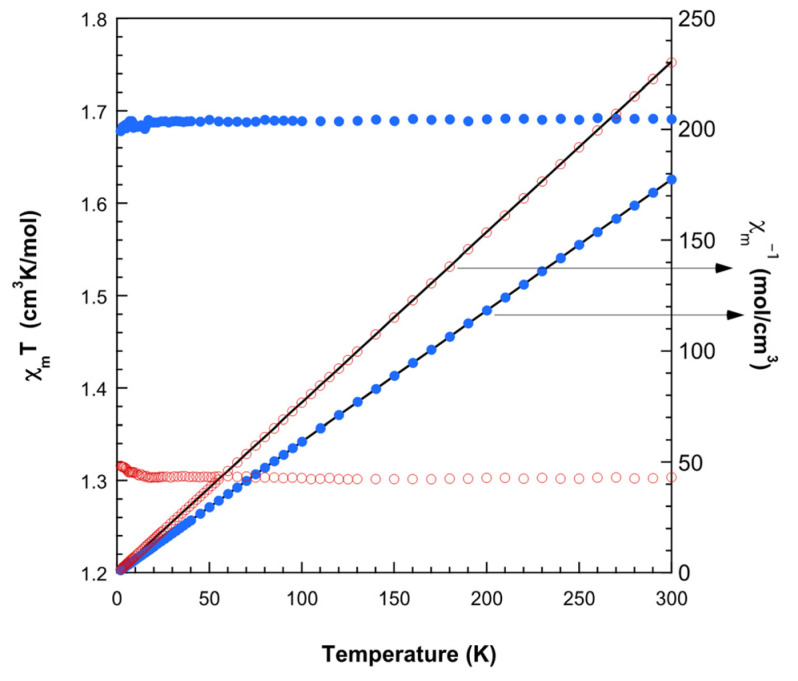
Magnetic behavior observed for **1** (open red circles) and **2** (solid blue circles). The Curie–Weiss fits are shown as solid lines on the reciprocal susceptibility data.

**Figure 12 ijms-25-13130-f012:**
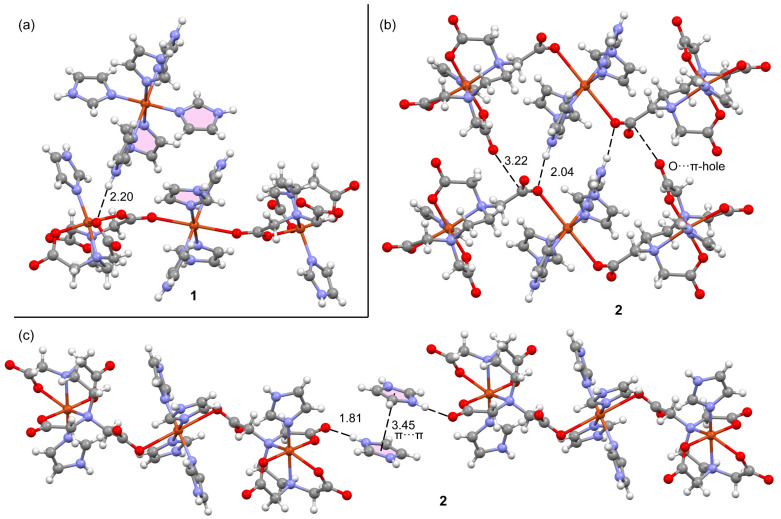
Partial views of the X-ray structures of compounds **1** (**a**) and **2**, with an indication of the self-assembled dimers of the trinuclear complex (**b**) and the s1D supramolecular polymer (**c**). Distances in Å. Color code: Cu, bronze; C, dark gray; O, red; N, blue; H, pale gray.

**Figure 13 ijms-25-13130-f013:**
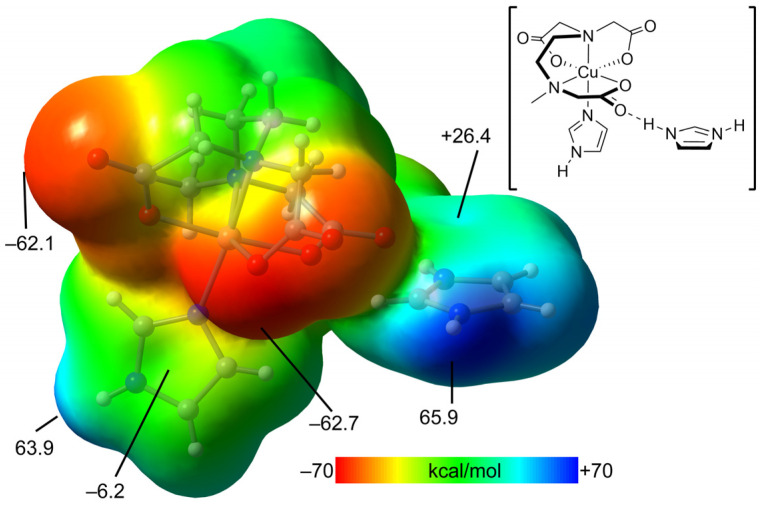
MEP surface of a reduced model of compound **2**. The MEP values at some points of the surfaces are given in kcal/mol. Level of theory: PBE0-D3/def2-TZVP. Isosurface 0.002 a.u.

**Figure 14 ijms-25-13130-f014:**
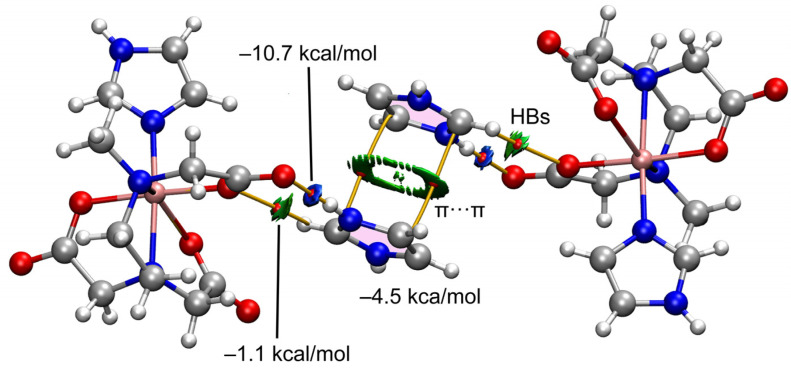
QTAIM/NCIPlot analysis of π-stacking in compound **2** at the PBE0-D3/def2-TZVP is shown. Only intermolecular contacts between the anionic unit and the electron donors are represented by bond CPs and RDG isosurfaces. See computational methods for the NCIplot settings. Color code: Cu, bronze; C, dark gray; O, red; N, blue; H, pale gray.

**Figure 15 ijms-25-13130-f015:**
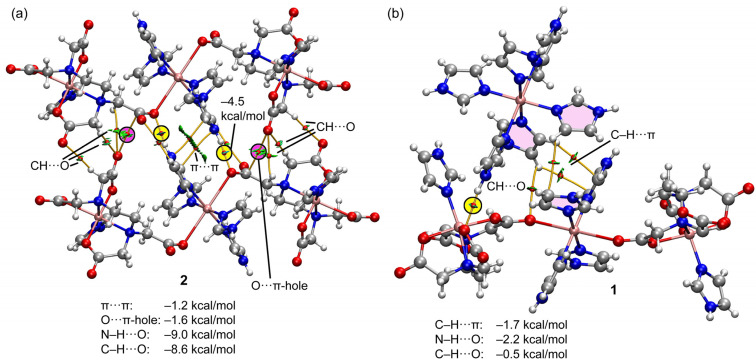
QTAIM/NCIPlot analyses of dimers of complexes **2** (**a**) and **1** (**b**) at the PBE0-D3/def2-TZVP are indicated. Only intermolecular contacts between the anionic unit and the electron donors are represented by bond CPs and RDG isosurfaces. See computational methods for the NCIplot settings. Color code: Cu, bronze; C, dark gray; O, red; N, blue; H, pale gray.

**Table 1 ijms-25-13130-t001:** Bond lengths and the shortest intermetallic distance (Å) with trans-coordination angles (°) in the linear trinuclear anion of compound 2 (see Figure 3).

Bond or Intermetallic	Distance (Å)	Trans-Angles (°)	
Cu(1)-O(4)	1.950(2)	O(4)-Cu(1)-O(15)	175.75(8)
Cu(1)-O(15)	1.974(2)		
Cu(1)-N(20)	1.988(2)	N(20)-Cu(1)-N(1)	168.44(9)
Cu(1)-N(1)	2.102(2)		
Cu(1)-N(12)	2.326(2)	N(12)-Cu(1)-O(8)	150.93(7)
Cu(1)-O(8)	2.456(2)		
Cu(2)-N(30)	1.996(2)	N(30)-Cu(2)-N(30)#1	180.0(1)
Cu(2)-N(30)#1	1.996(2)		
Cu(2)-N(25)	2.033(2)	N(25)-Cu(2)-N(25)#1	180.0
Cu(2)-N(25)#1	2.033(2)		
Cu(2)-O(19)	2.720(3)	O(19)#1-Cu(2)-O(19)	180.0
Cu(2)-O(19)#1	2.720(3)		
Cu(1)···Cu(2)	6.803(1)		

Symmetry code #1 = −x, −y + 2, −z.

**Table 2 ijms-25-13130-t002:** Thermogravimetric analyses (TGA) with identification of evolved gases by FT–IR library for compound **1**. MW: (1) = 1755.63, H_2_O = 18.015, CuO = 79.545.

Step or R	Temp. (°C)	Time (min)	Weight (%) Exp.	Weight (%) Cal.	Evolved Gases or Residue (R)
1	80–175	3–15	8.903	>6.157	6 H_2_O, CO_2_ (t)
2	175–260	15–24	28.909	-	CO_2_, H_2_O
3	260–270	24–26			CO_2_, H_2_O, CO
4	270–485	26–38	18.109	-	CO_2_, H_2_O, CO, N_2_O (t)
5	485–515	38–50	23.913	-	CO_2_, H_2_O, N_2_O, NO, NO_2_, X
R	565	57	18.341	18.123	CuO

t = trace amounts. X = non-identified gases. R = final solid residue(s).

**Table 3 ijms-25-13130-t003:** Thermogravimetric analyses (TGA) with identification of evolved gases by FT–IR library for compound **2**. MW: (1) = 1349.73, H_2_O = 18.015, CuO = 79.545.

Step or R	Temp. (°C)	Time (min)	Weight (%) Exp.	Weight (%) Cal.	Evolved Gases or Residue (R)
1	50–190	2–17	7.220	>>2.669	2 H_2_O (N), CO_2_
2	190–220	17–20	7.565	-	CO_2_, H_2_O
3	220–295	20–25	32.015		CO_2_, H_2_O, CO (t)
4	295–455	25–44	13.743		CO_2_, H_2_O, CO, N2O
5	455–500	44–47	13.771	-	CO_2_, H_2_O, CO, N_2_O, NO, CH_4_
6	500–580	47–60	7.767		CO_2_, H_2_O, CO, N_2_O, NO, NO_2_, CH_4_, X
R1	580	60	17.896	17.680	CuO (with some impurities)
R2	950	93	17.852	17.680	CuO

R = residue(s). N = undetermined value. t = trace amounts. X = non-identified gas(es).

## Data Availability

All data are available in the Appendix A.

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
