# Peer review of "Magnetic Isolation of the Linear Trinuclear Anion in [Cu(Him)6] {Cu(Him)4[Cu(μ-EDTA)(Him)]2}·6H2O (1) as the Novel Imidazolium(+) Salt (H2im)2[Cu(Him)4{(µ-EDTA)Cu(Him)}2]·2H2O (2)—A Comparative Look to Their Crystal Structures, Thermal, Spectral and Magnetic Properties and DFT Calculations"

_ijms, 2024, doi:10.3390/ijms252313130_

Round 1

Reviewer 1 Report

Comments and Suggestions for Authors

The paper deals with the synthesis and characterization of two linear trinuclear anions including besides [Cu(Him)6] {μ-Cu(Him)4[Cu(EDTA)(Him)]2}·6H2O (1) the novel imidazolium(+) salt (H2im)2[Cu(Him)4{(µ-EDTA)Cu(Him)}2]·2H2O (2). The Authors were inspired by the already reported crystal structure of compound 1 and succeeded in synthesizing and determining the structure of compound 2, where two imidazolium (H₂im⁺) ions serve as diamagnetic counter cations. The thermal stabilities, FT-IR, visible, and RSE spectra, as well as the magnetic properties of both compounds, together with DFT calculations, supporting the role of different interactions in stabilizing each crystal structure, are reported in the paper, which could be of some interest for the researchers in the field. Nevertheless, some key revisions are required  for a better outcome of the performed work.

1) In the introduction the Authors refer that “All studies include crystal structure determi-nations, such as a unique crystal that includes both mononuclear and cyclic trinuclear Cu(II) molecules [21]; (2) these compounds are of significant magnetic interest;……). The Authors should briefly describe the reasons of the “significant magnetic interest” with possible focus on potential applications.

2)Although in the experimental “green malachite Cu2CO3(OH)2 (the only basic carbonate of copper(II) with appropriate quality now commercially available)” is used as starting material in the previous results section basic copper(II) carbonate, is assumed to be and repeatedly reported as  Cu(OH)₂CO₃ . The Authors should clarify this point also justifying the stoichiometry in the latter case.

3) Since in the general synthetic procedure a specific warning reports: “Successive samples were checked by FT-IR to check whether samples correspond to the same phase (or not).” The Authors should better specify if the obtained product are the result of possible co-crystallization with other side products, indicating whenever possible the selective features of the isolated crystals of interest.  

Other typos such as “The heat is heating is interrupted,” line 453, should be corrected.

Author Response

The paper deals with the synthesis and characterization of two linear trinuclear anions including besides [Cu(Him)6] {μ-Cu(Him)4[Cu(EDTA)(Him)]2}·6H2O (1) the novel imidazolium(+) salt (H2im)2[Cu(Him)4{(µ-EDTA)Cu(Him)}2]·2H2O (2). The Authors were inspired by the already reported crystal structure of compound 1 and succeeded in synthesizing and determining the structure of compound 2, where two imidazolium (H₂im⁺) ions serve as diamagnetic counter cations. The thermal stabilities, FT-IR, visible, and RSE spectra, as well as the magnetic properties of both compounds, together with DFT calculations, supporting the role of different interactions in stabilizing each crystal structure, are reported in the paper, which could be of some interest for the researchers in the field. Nevertheless, some key revisions are required  for a better outcome of the performed work.

Response to the comment: We thank the reviewer for his/her positive comments

1)In the introduction the Authors refer that “All studies include crystal structure determinations, such as a unique crystal that includes both mononuclear and cyclic trinuclear Cu(II) molecules [21]; (2) these compounds are of significant magnetic interest;……). The Authors should briefly describe the reasons of the “significant magnetic interest” with possible focus on potential applications.

Response to the comment: The introduction has been modified describing the reasons of the “significant magnetic interest” with possible focus on potential applications.

2) Although in the experimental “green malachite Cu2CO3(OH)2 (the only basic carbonate of copper(II) with appropriate quality now commercially available)” is used as starting material in the previous results section basic copper(II) carbonate, is assumed to be and repeatedly reported as  Cu(OH)₂CO₃ . The Authors should clarify this point also justifying the stoichiometry in the latter case.

Response to the comment: This is an oversight on our part and the error has been corrected in the manuscript. Thank you for taking this to our attention. We have also clarified the stoichiometry by adding adjusted reactions to the experimental part

3) Since in the general synthetic procedure a specific warning reports: “Successive samples were checked by FT-IR to check whether samples correspond to the same phase (or not).” The Authors should better specify if the obtained product are the result of possible co-crystallization with other side products, indicating whenever possible the selective features of the isolated crystals of interest.

Response to the comment: Done as requested.

Other typos such as “The heat is heating is interrupted,” line 453, should be corrected.

Response to the comment: The errors were corrected.

Reviewer 2 Report

Comments and Suggestions for Authors

Regarding the description of the structures, it would be better to describe the octahedrons in the Cu environment with parameters that give a better idea of ​​the distortion.

You have some links to do it:

https://octadist.github.io/

https://www.degruyter.com/document/doi/10.1515/zkri-2018-2115/html

What is the relevance of Figure 5a? There is no reference to it in the text. In the plot, seems that gamma and beta are equals...

Some tables in S.I. need to be aligned.

Author Response

Comment: Regarding the description of the structures, it would be better to describe the octahedrons in the Cu environment with parameters that give a better idea of ​​the distortion.

You have some links to do it:

https://octadist.github.io/

https://www.degruyter.com/document/doi/10.1515/zkri-2018-2115/html

Response to the comment: We acknowledge and appreciate your documented comments. However, we believe that a study based on the quantitative treatment of the distortion of the octahedral coordination environment of the metal centers is not an appropriate approach for this particular manuscript, given its content. Furthermore, on page 2 (lines 46-50) of the manuscript, the typical distortion observed in a hexacoordinate environment in a Cu(II) ion is explained in detail.

Comment: What is the relevance of Figure 5a? There is no reference to it in the text. In the plot, seems that gamma and beta are equals...

Response to the comment: This is an oversight of our part and the error has been corrected in S.I. Figure 5a graphically explains the parameters collected in Table S6 of the S.I.

Comment: Some tables in S.I. need to be aligned.

Response to the comment: In S.I. the Tables have now been formatted.

Round 2

Reviewer 1 Report

Comments and Suggestions for Authors

The Authors have revised the manuscript according to the reviewer comments/suggestions